# Gastrodin Alleviates DSS-Induced Colitis in Mice through Strengthening Intestinal Barrier and Modulating Gut Microbiota

**DOI:** 10.3390/foods13152460

**Published:** 2024-08-03

**Authors:** Jiahui Li, Jinhui Jia, Yue Teng, Chunyuan Xie, Chunwei Li, Beiwei Zhu, Xiaodong Xia

**Affiliations:** 1State Key Laboratory of Marine Food Processing and Safety Control, National Engineering Research Center of Seafood, School of Food Science and Technology, Dalian Polytechnic University, 1 Qinggongyuan Road, Ganjingzi District, Dalian 116034, China; lijh@xy.dlpu.edu.cn (J.L.); jjh05271998@163.com (J.J.); foodsciyueteng@163.com (Y.T.); zhubeiwei@163.com (B.Z.); 2State Key Laboratory of Oncology in South China, Guangdong Provincial Clinical Research Center for Cancer, Sun Yat-sen University Cancer Center, Guangzhou 510060, China; xiecy@sysucc.org.cn; 3Guangdong Provincial Key Laboratory of Biomedical Imaging and Guangdong Provincial Engineering Research Center of Molecular Imaging, The Fifth Affiliated Hospital, Sun Yat-sen University, Zhuhai 519000, China; lichw35@mail2.sysu.edu.cn

**Keywords:** colitis, gastrodin, intestinal barriers, gut microbiota, inflammation

## Abstract

Inflammatory bowel diseases (IBDs) are commonly associated with dysfunctional intestinal barriers and disturbed gut microbiota. Gastrodin, a major bioactive ingredient of *Gastrodia elata* Blume, has been shown to exhibit anti-oxidation and anti-inflammation properties and could mitigate non-alcoholic fatty liver disease, but its role in modulating IBD remains elusive. The aim of this study was to investigate the impact of gastrodin on DSS-induced colitis in mice and explore its potential mechanisms. Gastrodin supplementation alleviated clinical symptoms such as weight loss, a shortened colon, and a high disease activity index. Meanwhile, gastrodin strengthened the intestinal barrier by increasing the 0expression of tight junction proteins and mucin. Furthermore, Gastrodin significantly reduced pro-inflammatory cytokine secretion in mice by downregulating the NF-κB and MAPK pathways. Gut microbiota analysis showed that gastrodin improved the DSS-disrupted microbiota of mice. These findings demonstrate that gastrodin could attenuate DSS-induced colitis by enhancing the intestinal barrier and modulating the gut microbiota, providing support for the development of a gastrodin-based strategy to prevent or combat IBD.

## 1. Introduction

Inflammatory bowel diseases (IBDs), including ulcerative colitis (UC) and Crohn’s disease (CD), are generally characterized by tissue damage and inflammation of the digestive tract [1]. The symptoms of IBD include ulceration, contraction of the intestines, diarrhea, blood in the stool, and changes in the gut microbiota [2]. Globally, the prevalence of IBD is estimated to have increased by 47% from 3.32 million in 1990 to 4.9 million in 2019, and the number of IBD cases is still on the rise, creating a significant burden on society and impacting quality of life [3]. UC differs from CD in that it presents with diffuse and continuous lesions, and inflammation generally affects the mucosa layer [4]. Currently, the main treatment strategies for IBD include corticosteroids, 5-aminosalicylic acid, immunomodulators, and antibodies (such as adalimumab) [5]. However, these drugs could cause some undesirable side effects [6,7]. Therefore, it is imperative to find an alternative strategy to combat IBD.

Many interrelated factors contribute to the pathogenesis of IBD, which include diet, microbiota, genetics, immune status, and environment [8]. Deciphering correlations between genetic locuses (such as NOD2 and IL-23R), various immune cells (such as macrophages and dendritic cells), cytokines, the intestinal microbiota, environmental factors (such as smoking and stress), and IBD could contribute to a better understanding of its pathogenesis. The compromised epithelial barrier has been shown to play an important role in IBD [9,10]. Various signaling pathways also contribute to the inflammatory process and are important in the development of IBDs, which include MAPK, NF-kappaB, and PI3K/Akt signaling pathways [11]. Moreover, it has been demonstrated that IBD is tightly associated with disturbances of the intestinal microbiota [12]. IBD patients have significantly different gut microbiotas than healthy individuals, including changes in the structure and function of the gut microbiota [13,14,15]. Increasing studies have demonstrated that modulation of the gut microbiome could effectively prevent or alleviate IBD [16,17].

Gastrodin, with a chemical structure of 4-hydroxybenzyl alcohol-4-O-β-D-glucopyranoside and a molecular weight of 286 Da, is the key bioactive ingredient in traditional Chinese medicine, *Gastrodia elata* Blume [18]. *Gastrodia elata* Blume has been traditionally utilized to alleviate headaches, dizziness, spasms, memory loss, and other conditions. Modern medical research has shown that gastrodin exhibits prominent antioxidant, anti-inflammatory, anti-apoptotic, and antiviral properties and has been used to treat neurological and vascular diseases [19,20,21]. As a main active component in a commonly used traditional Chinese medicine, gastrodin has the advantages of low toxicity (no observed toxicity at the dose of 5 g/kg bw in mice) and few side effects when it is leveraged to treat diseases [22]. Previous studies have shown that gastrodin could improve diabetic nephropathy induced by high glucose levels by alleviating inflammation and oxidative stress levels [23]. Another study indicates that gastrodin could restore prostaglandin E2 levels to alleviate ischemia/reperfusion and aspirin-induced double gastric mucosal damage, and also has an ameliorative effect on gastrointestinal dysfunction caused by atorvastatin [24,25]. In addition, gastrodin treatment significantly increased the relative abundance of probiotic bacteria such as *Lactobacillus*, *Bifidobacterium*, and *Bacteroidetes* and the levels of short-chain fatty acids (SCFA) such as butyric and isobutyric acids [26]. Although the above studies indicate that gastrodin has the potential to improve gastrointestinal damage and disorders, how gastrodin impacts IBD still remains elusive. In the present study, we reported that gastrodin could alleviate DSS-induced colitis in a mouse model through enhancing gut barrier function, rectifying gut microbiota, and attenuating inflammatory responses

## 2. Materials and Methods

### 2.1. Materials and Chemicals

Dextran Sodium Sulphate (DSS) at 98% purity was obtained from MP Biomedicals, LLC (MP Biomedicals, LLC, Santa Ana, CA, USA). Gastrodin was purchased from MCE (MedChemExpress LLC, Wuhan, China). The mouse ELISA (enzyme-linked immunosorbent assay) kits for TNFa, IL-6, IL-1β, and BCA were obtained from Thermo Fisher Scientific (Thermo Fisher Scientific, Waltham, MA, USA). Nitric oxide (NO) and myeloperoxidase (MPO) assay kits were bought from Nanjing Jiancheng Bioengineering Institute (Nanjing Jiancheng Biotechnology Research Institute Co., Ltd., Nanjing, China). Polyvinylidene fluoride film (PVDF) was purchased from Merck Chemicals (Merck & Co., Inc., Shanghai, China). A EDTA-free protease and phosphatase inhibitor cocktail (cOmplete^TM^) was purchased from Roche (Roche Pharma Schweiz Ltd., Penz-berg, Germany). EpiZyme Biotechnology (Shanghai Epizyme Biomedical Technology Co., Ltd., Shanghai, China) produced the SDS-PAGE sample loading buffer and DAPI staining solution. All other chemicals were of analytically pure grade (>99%).

### 2.2. Animals and Experimental Design

Twenty SPF and male C57BL/6 mice (8 weeks old) were purchased from Beijing Vital River Laboratory Animal Technology Co. (Beijing, China; Animal license no. SCXK [Jing] 2021-0006; Certification no. 110011231101033056). Animal experiments were approved by the Animal Experimentation Ethics Committee of Sun Yat-sen University Cancer Centre (Approval No. L102012021004U).

After one week of acclimatization, the mice were randomly divided into control group (*n* = 7), DSS intervention group (*n* = 6), and gastrodin intervention group (GAS + DSS) (*n* = 7). The mice in the control and DSS groups were gavaged with sterile distilled water, and the mice in the GAS + DSS group were gavaged with gastrodin (20 mg/kg/d) for the first two weeks. Gastrodin doses refer to the experimental method used by Wan et al. [27]. From the third week, mice in the DSS and GAS + DSS groups were given 2.5% DSS water freely for 1 week (Figure 1). During the experimental period, mice were monitored daily for body weight, food consumption, and other traits. At the end of the experiment, the mice were euthanized by asphyxiation using CO_2_, followed by exsanguination and major organ harvest. Blood, colon tissue, and cecum contents were collected from all mice and stored at −80 °C. Some of the colon tissues were fixed in 4% paraformaldehyde.

### 2.3. Histopathological Evaluation and PAS Staining

The fixed colon tissues were embedded in paraffin, sliced into 3–5 µm slices, and set aside. Changes in intestinal mucosal epithelial thickness and cup cells were observed using staining with hematoxylin and eosin (H&E) and periodic acid-Schiff stain (PAS). The histopathological features of the colon specimens were observed using a microscope (Nikon Eclipse Ti-S, Nikon, Tokyo, Japan), mainly including the degree of crypt disruption (Normal, 0; minor damage, 2; major damage, 4), the degree of inflammatory cell infiltration (Normal, 0; infiltration around crypts, 1; infiltration present in the muscular layer of the mucosa, 2; generalized infiltration in the muscular layer of the mucosa, 3; infiltration in the submucosal layer, 4) and the number of cup cells (0–2, 0; 2–4, 1; 4–6, 2; 6–8, 3; >8, 4).

### 2.4. Nitric Oxide (NO) Level and Myeloperoxidase (MPO) Activity Assay

The colon tissue was accurately weighed and homogenized in 0.1 M PBS (1:19 *w*/*v*, pH 7.4) to prepare a 5% tissue homogenate. The mixture was centrifuged, and the supernatant was collected. Colonic protein concentrations were determined using the BCA kit. MPO and NO levels were determined using the instructions provided by the manufacturer.

### 2.5. Inflammatory Cytokine Determination

Inflammatory cytokines in mouse colons were determined using ELISA kits. All measurements were performed following the manufacturer’s instructions. IL-1β, IL-6, and TNF-α were detected using an ELISA kit and calculated from the measured standard curves.

### 2.6. Quantitative Real-Time PCR Analysis

Total RNA was extracted using Trizol reagent (Shanghai Sangon Biotech Co., Ltd., Shanghai, China) according to the instructions, then quantified and reverse transcribed to cDNA using the Primer Script RT kit (Taraka Holdings Inc., Beijing, China) and genomic DNA annihilator. Next, samples were spiked using the ratio of SYBR premix kits (Tiangen Biotech Co., Ltd., Beijing, China), mixed, and analyzed by real-time PCR on a Bio-Rad detection system. The primers used were occludin, E-cadherin, muc2, tff3, IL-6, TNF-α, and IL-1β. Finally, the expression of individual genes was calculated based on ct values.

### 2.7. Western Blot Analysis

Frozen colon tissue was taken, and 9 times the volume of lysate (RIPA:PMSF:protease inhibitor:phosphatase inhibitor = 50:1:0.5:0.5) was added and homogenized in an ice-water bath, and left to stand for 30 min, with shaking every 15 min. The supernatant was centrifuged twice (1200× *g*, 15 min; 12,000× *g*, 15 min), and the quantified proteins were separated by polyacrylamide gel electrophoresis after the addition of supersampling buffer, after which they were transferred to PVDF membranes. The transferred PVDF membrane was closed using skimmed milk powder (5%) for 1 h at room temperature, washed clean with TBST, and incubated with primary antibodies (E-cadherin, occludin, p-P65, P65, p-JNK1/2/3, JNK1/2/3, p-ERK1/2, ERK1/2, p-P38, P38 and β-actin; Dilution ratio of 1:1000) overnight at 4 °C. On day 2, the primary antibody was aspirated, washed 3 times with tris buffered saline with Tween-20 (TBST), and incubated with the secondary antibody for an additional 1 h at room temperature. Then, the membrane was washed again with TBST and chemiluminescent substrate was added. Protein bands were then developed using a Bio-Rad image analysis system, and protein quantification was carried out using Image J software (Version 1.48b, US National Institutes of Health, Bethesda, MD, USA).

### 2.8. Immunofluorescence Assay

Briefly, fixed colon tissue sections were treated in xylene and then sequentially hydrated in graded ethanol for incubation of primary antibodies (E-cadherin and Occludin; Dilution ratio of 1:500) and secondary antibodies. The primary antibodies and FITC (fluorescein isothiocyanate)-labeled secondary antibodies used in this study. Photographs of the different antibody expressions were taken by fluorescence microscopy in the same area of the colon tissue.

### 2.9. Microbial Analysis

Colon contents were taken under sterile conditions, and all samples were snap-frozen at −80 °C for DNA extraction and analysis. Primers 341 F and 806 R were used to amplify the V3–V4 region of the bacterial 16S rRNA gene. Sequencing was conducted by Magigene Technology (Guangzhou, China), and Illumina Miseq PE250 (Illumina, San Diego, CA, USA) was used to generate raw data. After truncation and removal of primer information, further analysis was performed using the QIIIME2 platform. The sequences were denoised using the DADA2 algorithm to generate representative sequences and ASV tables. Finally, the representative bacterial gene sequences were classified and analyzed using the RDP Bayes-Classifier with a confidence threshold of 80%.

### 2.10. Statistical Analysis

Statistical analyses were performed with the GraphPad Prism software, V.8.0. One-way analysis of variance (ANOVA) was utilized for multiple comparisons and post hoc Tukey’s test for pairwise comparisons. The comparison of survival curves was calculated by the log-rank (Mantel–Cox) test in the GraphPad software. All data are presented as means ± SEM. A *p*-value of <0.05 was used for determining statistical significance. Data were expressed as mean ± SEM. (* *p* < 0.05, ** *p* < 0.01, *** *p* < 0.001, **** *p* < 0.0001).

## 3. Results

### 3.1. Gastrodin Relieved Clinical Symptoms Induced by DSS in Mice

Typical clinical symptoms that occur with colitis include weight loss, diarrhea, and bloody stools. Our experimental results showed obvious weight loss in mice in the DSS group. Surprisingly, Gastrodin pretreatment significantly attenuated DSS-induced weight loss (*p* < 0.001, Figure 2A,B). Moreover, mice in the gastrodin-treated group had lower disease activity index (DAI) scores (*p* < 0.05, Figure 2C) and longer colon lengths (*p* < 0.05, Figure 2D,E) compared to those in the DSS group. These results suggest that gastrodin can effectively attenuate DSS-induced colitis in mice.

### 3.2. Gastrodin Ameliorates Histopathological Changes in the Colon of Mice

Compared with the control group, mice in the DSS group suffered severe colonic injury, the structure of the crypts was severely damaged and significantly increased the infiltration of inflammatory cells infiltrations, while supplementation of gastrodin reversed these changes (Figure 3A,B). PAS staining results showed that DSS led to a decrease in the number of goblet cells in the crypts and a decrease in the level of mucin in the colonic tissues. Gastrodin intervention increased mucin levels and reduced mucus layer damage (Figure 3B,D). In addition, the gastrodin group had lower histological scores compared to than the DSS group (*p* < 0.01, Figure 3C). Taken together, these results suggest that gastrodin could attenuate DSS-induced histopathologic damage in the colon.

### 3.3. Gastrodin Increased the Expression of Epithelial Tissue Protein in Mice

Compared to the control group, the expression levels of occludin, E-cadherin, muc2, and tff3 mRNA were significantly reduced in the colonic tissues of mice in the DSS group, while enhanced expression of these genes was observed in mice in the gastrodin-treated group (Figure 4A–D). To further validate the effect of DSS on the intestinal barrier, we used Western blotting and immunofluorescence to detect the expression of occludin and E-cadherin in tissues. As shown in Figure 4E–H, occludin and E-cadherin were almost absent in the DSS-treated group, which was also confirmed by immunofluorescence assay, whereas gastrodin treatment significantly elevated the expression of occludin and E-cadherin in the colon. The above results suggest that gastrodin may ameliorate DSS-induced intestinal injury by repairing the intestinal barrier function. 

### 3.4. Gastrodin-Modulated Inflammatory Cytokines and Oxidative Stress in Mice

A marker of colonic injury is the excessive expression of pro-inflammatory cytokines. The levels of pro-inflammatory cytokines such as IL-6, TNF-α, and IL-1β in intestinal tissues were significantly upregulated in the DSS group compared to the control group, whereas gastrodin supplementation significantly down-regulated the levels of pro-inflammatory cytokines (*p* < 0.05, Figure 5A–C). RT-PCR showed that the gastrodin-intervened group showed decreased transcriptional expression of IL-6, TNF-α, and IL-1β (*p* < 0.001, Figure 5D–F). In addition, we also measured indicators of oxidative stress, which is closely related to inflammation. MPO is an enzyme that connects the inflammatory response to oxidative stress and catalyzes oxidative reactions, which produce large amounts of cytotoxic reactive oxygen species. As shown in Figure 5G, gastrodin supplementation significantly reduced colonic MPO levels. NO levels are thought to be strongly associated with disease activity in ulcerative colitis. As depicted in Figure 5H, gastrodin significantly reduced NO levels in DSS-induced colitis mice and improved the oxidative stress and inflammatory status of the organism compared with the DSS group.

### 3.5. Gastrodin Suppresses Inflammatory Response through Modulating NF-κB and MAPK Pathways

Abnormal activation of the NF-κB and MAPK signaling pathways is considered to be an important mechanism of IBD pathogenesis. Compared with the control group, the phosphorylation levels of NF-κB p65 and key proteins on the MAPK pathway (JNK, ERK, p38) were significantly higher in the DSS group, suggesting that the intervention of DSS led to the activation of NF-κB and MAPK signaling pathways. Gastrodin treatment significantly downregulated the protein levels of NF-κB p65 and MAPK-related proteins. These data suggest that gastrodin could regulate inflammation partly through the NF-κB and MAPK pathways (Figure 6). 

### 3.6. Gastrodin-Modulated Intestinal Microbiota in DSS-Induced Colitis Mice

The α-diversity index reflected the differences in microbial communities among the groups. As shown in Figure 7A, the Simpson index was significantly lower in mice in the DSS group compared with that of the control group, indicating that DSS treatment led to the destruction of the species diversity of the mouse flora (*p* < 0.01). Gastrodin treatment could reverse the decrease in the diversity of the intestinal microbiota to a certain extent (*p* < 0.05). The PCA clustering analysis showed that the intestinal microbial communities of the DSS-treated mice were separated from those of the control group mice and that the gastrodin-treated mice were clustered closer to the control mice (Figure 7B). Analyses at the phylum level showed that, compared to that in the control group, the relative abundance of Firmicutes was lower in the DSS group, whereas the relative abundance of Bacteroidetes was higher, further suggesting that the addition of DSS led to the disruption of the intestinal flora of the mice. (Figure 7C–E). The relative abundance of Proteobacteria was low in the control group, while significantly increasing in the DSS group significantly increased the relative abundance of Proteobacteria. Gastrodin supplementation led to a decrease in the relative abundance of Proteobacteria without statistical significance (Figure 7F). In addition, at the genus level, gastrodin supplementation up-regulated strains of Laetobaeillus, Turicibacter, and Parabacteroides and down-regulated strains of Bacteroides, Akkermansia, and Escherichia-Shigella (Figure 7G,H). 

## 4. Discussion

Increasing amounts of traditional herbs have been examined for their potential to prevent or mitigate inflammatory diseases such as IBD [28]. Gastrodin is the main ingredient in *Gastrodia elata*, which is widely used in China to treat epilepsy, headaches, paralysis, and dizziness [18,29]. However, how gastrodin impacts IBD and its potential mechanisms have yet to be determined. Herein, our study revealed that gastrodin alleviated colitis in mice partly through enhancing the gut epithelial barrier and modulating the gut microbiota.

Compromised intestinal barrier integrity is one of the main characteristics of inflammatory bowel diseases. Tight junction proteins (TJs) are mainly found between intestinal epithelial cells and play an important role in maintaining the integrity of the intestinal barrier. TJs include cytosolic scaffold proteins (AF-6, zonulae occludens [ZO] 1–3, and cingulin) and transmembrane proteins (occludin, claudins, tricellulin, and junctional adhesion molecules [JAMs]) [30]. Goblet cells are an important intestinal epithelial cell that regulates mucus secretion and wound healing by secreting trefoil factor 3 (TFF3) and MUC2. TFF3 has been implicated in intestinal mucosal injury and repair. It has been shown that mice lacking the TFF3 gene are particularly sensitive to DSS-induced mucosal injury, and intestinal epithelial regeneration is severely deficient, leading to massive mortality in mice with colitis [31]. The mucus layer consists of core proteins such as mucin-2, secreted by goblet cells, and other secreted substances. Meanwhile, impaired mucus layer protection is one of the main causes of infection, various chronic inflammatory diseases, and intestinal bacterial invasion [32]. In addition, the number of goblet cells and mucin expression are important factors influencing the intestinal barrier. E-cadherin is a cell adhesion protein that is a key substance in maintaining the structural and functional integrity of epithelial tissues and regulating intercellular adhesion. Inactivation of E-cadherin changed the maturation of goblet cells, increased epithelial cell apoptosis and shedding, reduced AMP release, and caused bloody diarrhea [33]. In our study, the DSS group mice showed increased intestinal permeability and reduced expression of E-cadherin, occludin, TFF3, and MUC2. Nevertheless, gastrodin supplementation could effectively up-regulate the expression of these junctional proteins, reversing DSS induced intestinal integrity damage, suggesting that gastrodin has a beneficial effect on intestinal barrier function.

Several studies have shown that the pathogenesis of IBD is related to the aberrant upregulation of the NF-κB signaling pathway and the MAPK signaling pathway. During intestinal injury, NF-κB activation leads to the production of a series of pro-inflammatory cytokines, resulting in impaired intestinal integrity. MAPK disrupts the balance of pro-inflammatory and anti-inflammatory factors. It has been shown that gastrodin can down-regulate NF-κB expression, which inhibits the downstream cascade response, leading to a decrease in pro-inflammatory cytokine secretion [34]. Gastrodin also reduces the secretion of pro-inflammatory cytokines and ameliorates acetaminophen (APAP)-induced liver injury in mice by inhibiting the MAPK signaling pathway [35]. In our study, gastrodin effectively inhibited the activation of NF-κB and MAPK signaling pathways and improved the inflammatory response. These results suggest that gastrodin may exert its anti-colitis effect through dephosphorylation of inflammatory signaling pathways and a reduction in IL-6, TNF-α, and IL-1β levels.

In the development of IBD, pro-inflammatory cytokines could activate immune cells, causing increased inflammation and ROS production [36]. Colitis is associated with oxygen-free radical formation and oxidative stress. MPO levels were significantly elevated in DSS-induced mouse colon tissues. However, gastrodin significantly reduced MPO activity. In addition, the production of iNOS-derived nitric oxide (NO), whose production is closely associated with the progression of ulcerative colitis. Gastrodin significantly reduced NO levels in the colonic mucosa of mice, indicating that it may ameliorate colitis partly by modulating oxidative stress.

DSS treatment led to a significant reduction in the diversity of the intestinal flora, disrupting the original flora structure. This change was not only reflected in the decrease in the number of certain beneficial bacterial species but also included a relative increase in the number of harmful bacterial species, which further deteriorated the intestinal environment [34]. Our study showed that gastrodin ameliorates the microbiota disturbance induced by DSS in mice. The beneficial effects of gastrodin on gut flora have been reported before. For example, gastrodin may regulate intestinal health by increasing the abundance of beneficial bacteria and decreasing the abundance of pathogenic bacteria [37]. DSS-induced disruption of the intestinal flora is not limited to a direct reduction in the diversity of the flora. Deeper effects are seen in changes in the structure of the flora. For example, changes in the relative abundance of major phylums such as Firmicutes and Bacteroidetes are altered, which have a direct impact on the gut immune response and the degree of inflammation. Some of the beneficial species in Firmicutes can modulate the host immune response through the production of metabolites such as short-chain fatty acids, whereas an increase in Bacteroidetes may be associated with increased intestinal inflammation and pathological damage [26,38]. Gastrodin reversed the DSS-induced decrease in the abundance of Firmicutes and the increase in the abundance of Bacteroidetes, ameliorating the disruption of the intestinal flora structure. In addition, DSS-induced intestinal flora disruption is often accompanied by an increase in Proteobacteria, which are considered to be harmful bacteria, and their increased abundance may lead to increased intestinal inflammation [39]. In addition, gastrodin upregulated the abundance of beneficial bacteria such as *Laetobaeillus*, Turicibacter, and Parabacteroides, and decreased that of Bacteroides, Akker-mansia, and Escherichia-Shigella, which have been reported to have positive effects on intestinal health with a regulatory role with a moderating effect, have been reported to have significant effects on intestinal health with significant regulatory effects [40,41,42,43]. However, gastrodin-treated mice showed a decreasing trend for the relative abundance of Proteobacteria, demonstrating its ability to reduce deleterious bacterial taxa.

## 5. Conclusions

In summary, our results suggest that gastrodin could significantly ameliorate colitis in mice by enhancing the expression of mucin, E-cadherin, and tight junction protein occludin, attenuating the inflammation-related NF-κB signaling pathway, alleviating oxidative stress, and inhibiting the expression of pro-inflammatory cytokines including TNF-α, IL-1β, and IL-6. In addition, the beneficial effects of gastrodin on colitis were also associated with the modulation of the gut microbiota, as evidenced by the increased diversity of bacterial species and the up-regulation of the abundance of beneficial bacteria such as *Lactobacillus*. These findings indicate that gastrodin may be used as a complementary strategy for the treatment of IBD. However, further studies are needed to elucidate the specific mechanisms before practical application.

## Figures and Tables

**Figure 1 foods-13-02460-f001:**
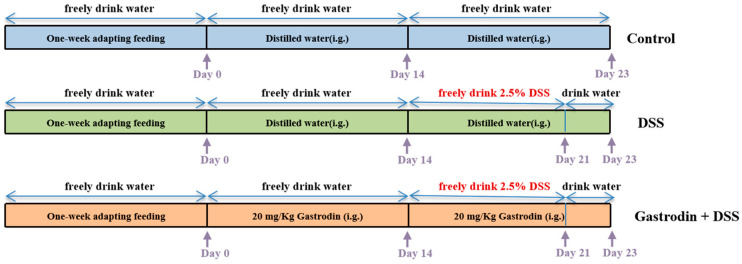
Schematic representation of the experimental design.

**Figure 2 foods-13-02460-f002:**
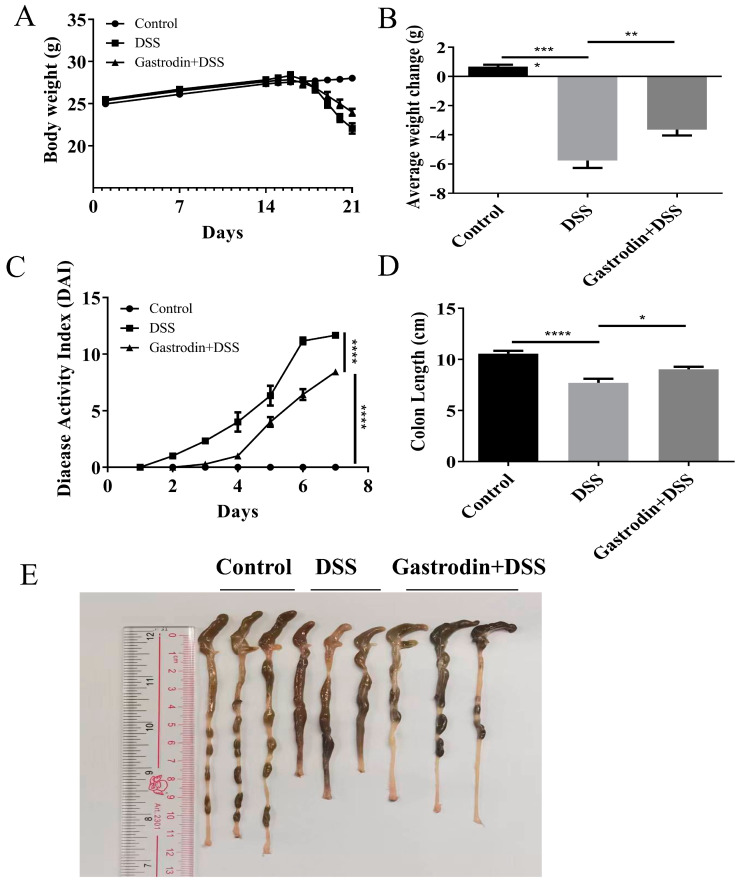
Gastrodin supplementation relieved DSS-induced colitis in mice. (**A**) Body weight of mice in different groups (*n* = 6–7). (**B**) Average body weight change at the time of sacrifice. (**C**) Disease activity index. (**D**) Colon length (cm). (**E**) Representative images of the colon. Data are presented as mean ± SEM. * *p* < 0.05, ** *p* < 0.01, *** *p* < 0.001, **** *p* < 0.0001.

**Figure 3 foods-13-02460-f003:**
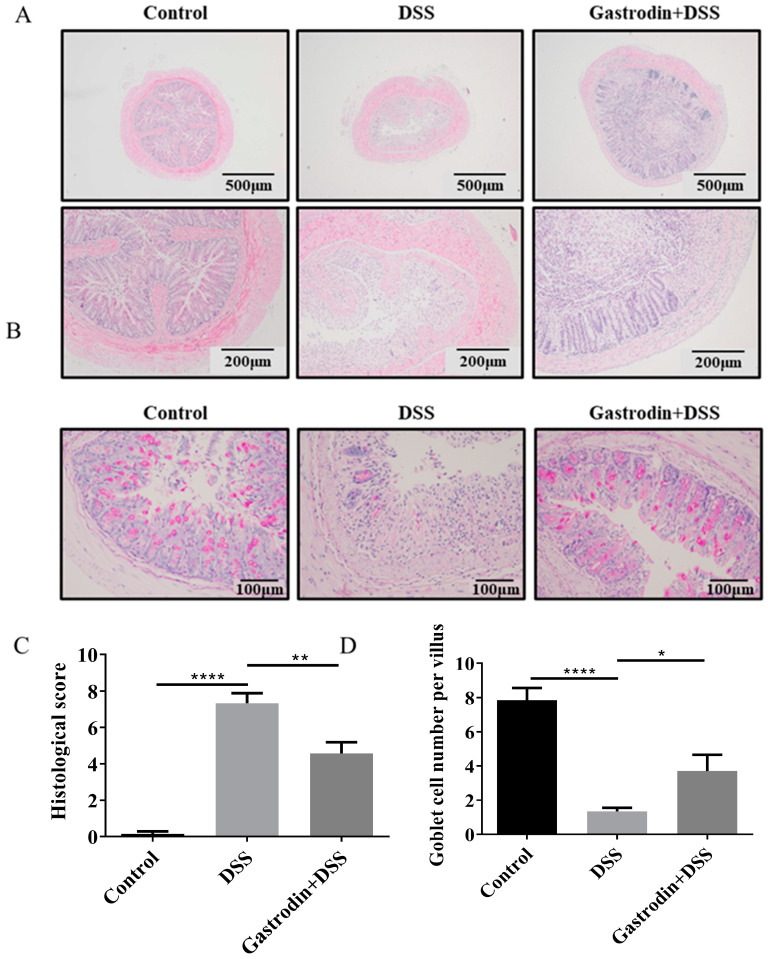
Gastrodin improved histopathological alterations in colitis mice. Colonic tissues were subjected to (**A**) HE staining and (**B**) PAS staining. (**C**) Histological score (*n* = 6–7) of HE-stained sections and (**D**) the number of villous cells per villus (*n* = 6–7). Data are presented as mean ± SEM. * *p* < 0.05, ** *p* < 0.01, **** *p* < 0.0001.

**Figure 4 foods-13-02460-f004:**
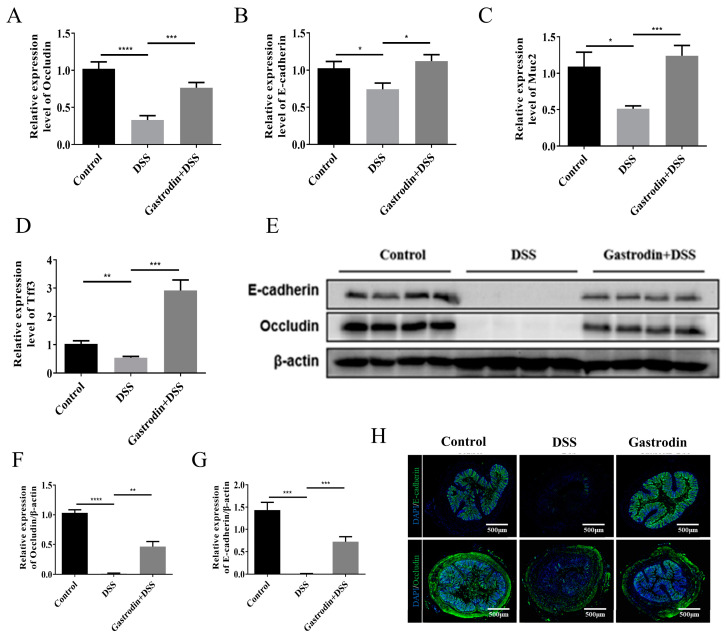
Gastrodin strengthened the intestinal barrier in mice with DSS-induced colitis. Relative mRNA expression of occludin (**A**), E-cadherin (**B**), muc2 (**C**), and tff3 (**D**) in the colon tissue (*n* = 6–7). (**E**) Western blot analysis of E-cadherin and occludin proteins and relative quantification of occludin (**F**) and E-cadherin (**G**) proteins (*n* = 4). (**H**) Immunostaining of E-cadherin, and Occludin. Magnification: 40×. Data are presented as mean ± SEM. * *p* < 0.05, ** *p* < 0.01, *** *p* < 0.001, **** *p* < 0.0001.

**Figure 5 foods-13-02460-f005:**
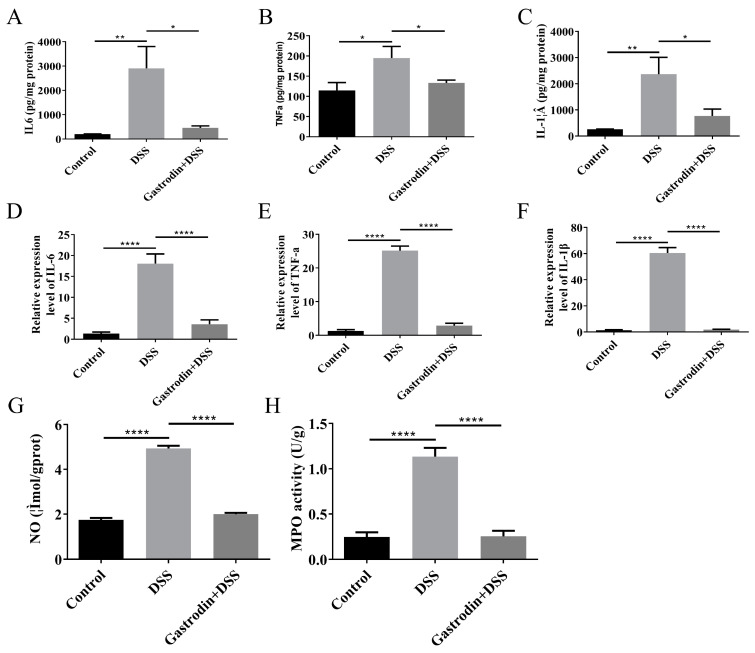
Gastrodin reduces inflammatory cytokines and oxidative stress levels in mice with colitis. (**A**) IL-6, (**B**) TNF-α, and (**C**) IL-1β levels of intestinal tissues were measured by ELISA (*n* = 6–7). (**D**) IL-6, (**E**) TNF-α, and (**F**) IL-1β mRNA expressions were measured by RT-PCR (*n* = 6–7). (**G**) NO, (**H**) MPO levels in the colon (*n* = 7). Data are presented as mean ± SEM. * *p* < 0.05, ** *p* < 0.01, **** *p* < 0.0001.

**Figure 6 foods-13-02460-f006:**
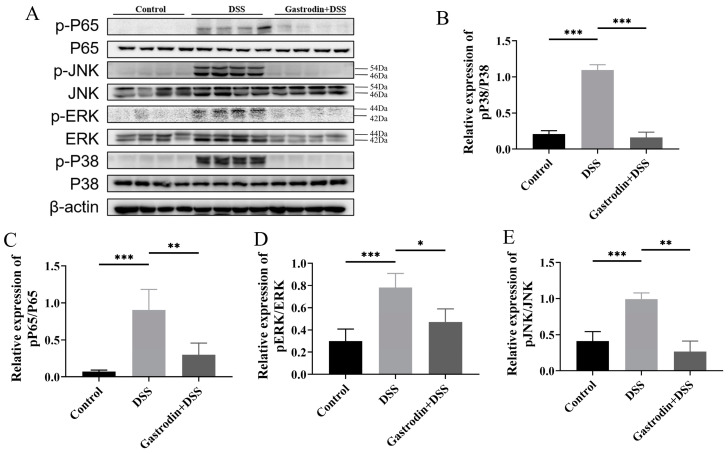
Gastrodin-inhibited NF-κB and MAPK pathways in mice with DSS-induced colitis. (**A**) Western-blot analysis of p-P65, P65, p-JNK, JNK, p-ERK, ERK, p-P38, and P38 proteins and (**B**–**E**) relative quantifications of those proteins by densitometry. Data are presented as mean ± SEM. * *p* < 0.05, ** *p* < 0.01, *** *p* < 0.001.

**Figure 7 foods-13-02460-f007:**
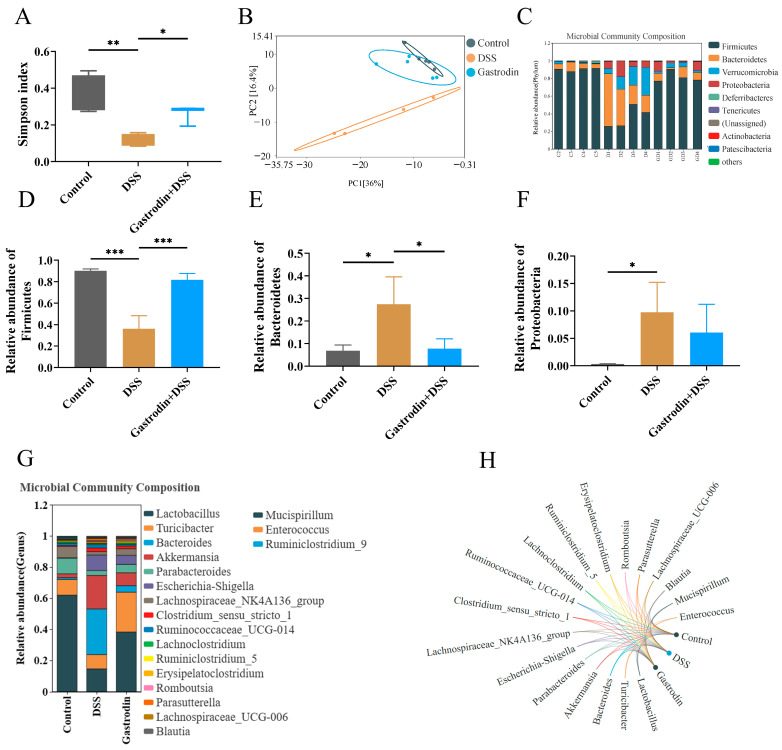
Gastrodin improved disturbed gut microbiota induced by DSS in mice. (**A**) Simpson index was assessed (**B**) Principal coordinate analysis (PCoA) of weighted UniFrac distance was conducted for microbiota of mice from different groups. (**C**) Relative abundance of operational taxonomic units (OTUs) at the phylum level. (**D**) Relative abundance of Firmicutes, (**E**) Bacteroidetes, and (**F**) Proteobacteria in mice from different groups. (**G**) Relative abundance of flora at the Genus level, and (**H**) Correlation network analysis at the genus level. Data are presented as mean ± SEM. * *p* < 0.05, ** *p* < 0.01, *** *p* < 0.001.

## Data Availability

The original contributions presented in the study are included in the article, further inquiries can be directed to the corresponding author.

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
