# Peer review of "Gastrodin Alleviates DSS-Induced Colitis in Mice through Strengthening Intestinal Barrier and Modulating Gut Microbiota"

_foods, 2024, doi:10.3390/foods13152460_

Round 1

Reviewer 1 Report

Comments and Suggestions for Authors

The authors have reported the potential of Gastrodin in alleviating DSS-induced colitis in mice through strengthening intestinal barrier and modulating gut microbiota. The manuscript adds value to the existing knowledge by exploring its role in IBD along with its mechanism. However, the quality of manuscript can be improved by addressing the following comments in the revised version;

1. It is not mentioned why did the authors have choose gastrodin for this study? A rationale is needed. Does Gastrodia elata Blume, used traditionally in TCM or elsewhere to treat IBD? Because, there are so many natural products possessing antioxidant and anti-inflammatory activities but they might not be beneficial in IBD. Hence, more justification or preliminary data is required for undertaking the study.

2. The paragraph 3 of the introduction section provide background information on gastrodin. However, it is too short. It would be appropriate to include its chemical structure and previous studies related to IBD, if any. Besides, mention in one or two sentences about its toxicity as well. 

3. Abbreviations must be expanded at their first place of occurrence  in the text. There are typos and grammatical errors in the manuscript, hence, it needs thorough revision. 

4. In line 87 and 88, GAS is written as Gas, plant name, where ever mentioned, should be italic. Check line 62, "whether an how gastrodin... In line 110, ELISA is written as elisa. Line 270, check spelling of modulating, line 325, before, etc. 

5. Sec 2.4. line 103-104: It looks like a heading; Determination of colonic tissue protein concentration using BCA kit. 

6. In all figures, it is written; Data were presented .... This should be in present tense, ie Data are presented....

7. On what basis the dose of gastrodin was decided? If taken a literature, cite a suitable  reference? Was any acute toxicity done?

8. Include correlation network analysis of microbial genus levels between Gastrodin and control  and DSS group. 

9. Elaborate conclusion. 

Comments on the Quality of English Language

There are typos/grammatical errors. English language needs to be thoroughly revised. 

Author Response

Reviewer Comments:

The authors have reported the potential of Gastrodin in alleviating DSS-induced colitis in mice through strengthening intestinal barrier and modulating gut microbiota. The manuscript adds value to the existing knowledge by exploring its role in IBD along with its mechanism. However, the quality of manuscript can be improved by addressing the following comments in the revised version:

  1. It is not mentioned why did the authors have choose gastrodin for this study? A rationale is needed. Does Gastrodia elata Blume, used traditionally in TCM or elsewhere to treat IBD? Because, there are so many natural products possessing antioxidant and anti-inflammatory activities but they might not be beneficial in IBD. Hence, more justification or preliminary data is required for undertaking the study.

Response: Thank you for your suggestion, we have added some background information to justify the necessity of the current work. Although gastrodin has been well known for its neuroprotective effects, some studies also suggest that it may alleviate gastrointestinal disorders. But currently whether and how it could impact IBD has not been examined. Therefore we designed this study to fill this gap of knowledge. We have added informaition in the introduction as follows: Another study indicates that gastrodin could restore prostaglandin E2 levels to alleviate ischemia/reperfusion and aspirin-induced double gastric mucosal damage, and also has an ameliorative effect on gastrointestinal dysfunction caused by atorvastatin [24,25]. In addition, gastrodin treatment significantly increased the relative abundance of probiotic bacteria such as Lactobacillus, Bifidobacterium, and Bacteroidetes and the levels of short-chain fatty acids (SCFA) such as butyric and isobutyric acids [26]. Although the above studies indicate that gastrodin has potential to improve gastrointestinal damage and disorders, how gastrodin impacts IBD still remain elusive.

  1. The paragraph 3 of the introduction section provide background information on gastrodin. However, it is too short. It would be appropriate to include its chemical structure and previous studies related to IBD, if any. Besides, mention in one or two sentences about its toxicity as well.

Response: Thank you for your suggestion, based on your suggestion we have revised and added the information as follows: Gastrodin, with a chemical structure of 4-hydroxybenzyl alcohol-4-O-β-D-glucopyranoside and a molecular weight of 286 Da, is the key bioactive ingredient in the traditional Chinese medicine, Gastrodia elata Blume. Gastrodia elata Blume has been traditionally utilized to alleviate headaches, dizziness, spasms, memory loss and other conditions. Modern medical research has shown that gastrodin  exhibit prominant antioxidant, anti-inflammatory, anti-apoptotic and antiviral properties and has been used to treat neurological and vascular diseases. As a main active component in a commonly used traditional Chinese medicine, gastrodin has advantages of low toxicity (no observed toxicity at the dose of 5g/kg bw in mice) and few side effects when it is leveraged to treat diseases

  1. Abbreviations must be expanded at their first place of occurrence in the text. There are typos and grammatical errors in the manuscript, hence, it needs thorough revision.

Response: We have expanded the abbreviations when they first appear in the text. And we used MDPI language service to help revise the language issues.

  1. In line 87 and 88, GAS is written as Gas, plant name, where ever mentioned, should be italic. Check line 62, "whether an how gastrodin... In line 110, ELISA is written as elisa. Line 270, check spelling of modulating, line 325, before, etc.

Response: Thank you for your suggestions. We have made corresponding corrections in the revised manuscript.

  1. Sec 2.4. line 103-104: It looks like a heading; Determination of colonic tissue protein concentration using BCA kit.

Response: Thank you for your suggestions, we have made changes in the manuscript.

  1. In all figures, it is written; Data were presented .... This should be in present tense, ie Data are presented....

Response: We have changed to present tense as suggested.

  1. On what basis the dose of gastrodin was decided? If taken a literature, cite a suitable reference? Was any acute toxicity done?

Response. The dose of gastrodin was determined by reference to two previous studies[1][2]. Since acute toxicity of gastrodin has been assessed in mice in several previous studies (no observed toxicity at a dose up to 5mg/kg bw in mice) [3], we did not conduct acute toxicity tests in this study.

[1].  Wan J, Zhang Y; Yang D.; Liang Y; Yang L.; Hu S, Liu, Z., Fang Q; Tian, S.; Ding Y. Gastrodin Improves Nonalcoholic Fatty Liver Disease Through Activation of the Adenosine Monophosphate–Activated Protein Kinase Signaling Pathway. Hepatology 2021.

[2].  Ma, S. , Sun, Y. , Zheng, X. , and Yang, Y. Gastrodin attenuates perfluorooctanoic acid-induced liver injury by regulating gut microbiota composition in mice. Bioengineered, 2021. 12(2), 11546-11556..

[3] Yun-Jiang M , Shi-Xian D .Pharmacological studies on gastrodia elata blume. [J].Acta Botanica Yunnanica, 1980.

  1. Include correlation network analysis of microbial genus levels between Gastrodin and control and DSS group.

Response: Thanks. We have performed correlation network analysis as suggested and added the results to the Figure 7H in the revised manuscript.

  1. Elaborate conclusion.

Response: We have elaborated the conclusion as follows: In summary, our results suggest that gastrodin could significantly ameliorate coli-tis in mice through enhancing the expression of mucin, E-cadherin and tight junction protein occludin, attenuating inflammation-related NF-κB signaling pathway, allevi-ating oxidative stress, and inhibiting the expression of pro-inflammatory cytokines in-cluding TNF-α, IL-1β, and IL-6. In addition, the beneficial effects of gastrodin on colitis were also associated with the modulation of the gut microbiota, as evidenced by in-creased diversity of bacterial species and up-regulation of the abundance of beneficial bacteria such as Lactobacillus.These findings indicate that gastrodin may be used as a complementary strategy for the treatment of IBD. However, further studies are needed to elucidate the specific mechanisms before practical application.

Reviewer 2 Report

Comments and Suggestions for Authors

Comments are listed in the file attached below 

Comments on the Quality of English Language

Major revision of grammar, punctuation and syntax is needed.  

Author Response

Comments:

The paper demonstrates the beneficial effects of gastrodin in alleviating DSS-induced colitis in mice. The article provides relevant results and new insight into the use of gastrodin for strengthening the intestinal barrier also to modulate the composition of gut microbiota.

However, I have some concerns outlined in the comments below.

  1. The authors have wrien that the numbers of IBD cases is on the rise imposing a burdern on the society, but do not write any specific data to let the reader comprehend the impact of this number. I suggest adding some epidemiologic data.

Response: Thank you for your suggestions. We have added epidemiologic data in the revised introduction as suggested: Globally, the prevalence of IBD is estimated to have increased by 47% from 3.32 million in 1990 to 4.9 million in 2019, and the number of IBD cases is still on the rise, creating a significant burden on society and impacting quality of life.

  1. The introduction lacks clarity and structure, the sentences appear somewhat disconnected from each other and the information appears superficial and lacking in depth. it might, for example, be useful to include which more specific information about the mechanisms that lead to the conditions mentioned.

Response: We have revised the introduction to make it more coherent. And some mechanisms were provided

  1. The botanical names must be wrien in italics.

Response: We have italicized all the botanical names as suggested.

  1. The authors do not indicate how the concentration of gastrodin used for treatment was chosen; a paragraph about this should be added. Were preliminary in vitro tests done? Or were they derived from literature research?

Response: The dose of gastrodin was determined with reference to two previous studies in which gastrodin at the same dose was used to treat hepatic disorders: "Gastrodin attenuates perfluorooctanoic acid-induced hepatic injury by modulating the composition of the intestinal microbiota in mice", and “Gastrodin Improves Nonalcoholic Fatty Liver Disease Through Activation of the Adenosine Monophosphate–Activated Protein Kinase Signaling Pathway” .

  1. Punctuation and syntax should be revised as well as grammar.

Response: Thanks. And we have used MDPI language service to help revise the language issues.

  1. Why was the oral gavage technique used for treatment administration? To date, alternative, noninvasive methods such as micropipette-guided drug administration or administration via food or water are recommended as these methods increase animal welfare.

Response: Thanks for your suggestion. At the time of experiments, we referred to previous similar studies and used oral gavage as a means of administration to control the exact amount of gastrodin given to each mouse. In our future studies, we will consider use non-invasive methods or supplement gastrodin in food or water to increase animal welfare and mimic natural intake in human consumption.

  1. Figure 2 is named in the text before figure 1.

Response: Thank you for your suggestions, we have corrected the order in the revised manuscript..

  1. The purpose of the work should be better explained. Is gastrodin administered for preventive purposes or for treatment purposes? This is not made explicit well in the text.

Response: Thank you for your suggestion, this experiment was designed to explore the preventive effect of gastrodin on colitis by administering a gastrodin intervention two weeks prior to modeling with DSS. We have made this clear in the revised manuscript as suggested.

  1. It is not stated how the rats were euthanized; I find this an important detail to make transparent.

Response: Thanks to your suggestion. The mice were euthanized by asphyxiation using CO2.

  1. The methods used for histopathological evaluation should be better described with an explanation of the protocol used.

Response: Thanks for your suggestions, we have described in detail the protocol used and cited the reference.

  1. Paragraph 2.4 is unclear; it should be better written. This also applies to paragraph 2.7.

Response: We have added more details to sections 2.4 and 2.7 to make them easier to understand.

  1. In section 2.5, cell cultures never mentioned before are named. Where do they emerge from?

Response: Sorry for the mistake..We have removed the sentence in the revised manuscript.

  1. In sections 2.7 and 2.8, I suggest listing the antibodies used for the experiments, so as to make the experimental protocol followed immediately clear. Also, in section 2.6 should be mentioned the gene analyzed.

Response: Thank you very much for your suggestion, we have added the information of antibodies used and genes examined.

  1. In Figure 2A, the DSS and gastrodin+DSS groups are not statistically significant compared to the control in the last time frame? Especially considering that in Figure 2B there is statistical significance.

Response: Sorry we forgot to add the significance symbol in the Figure 2A previously, now we have corrected it.

  1. Lines 184-187 are not relevant to the results section; this part should be included in the introduction or in support of the discussion.

Response: Thank you for your suggestion, we have moved it to the discussion part.

  1. The title of section 3.3 indicates the study of TJs alone and so does the introduction part of this section, however, proteins that are not TJs but are rather involved in their regulation are then described. In addition, it would also be appropriate to analyze a member of the claudin family, as they are major components of TJs with occludins.

Response: Thanks. We have revised it to barrier-related proteins to make the title more inclusive. It is a caveat of this study that claudins were not examined here and we will bear this in mind and include this tight junction protein in our future studies. 

Further, the graphs in this section should indicate the unit of measurement on the yaxis, as done in Figure 5 (A-C) 17. In line 215, reference 23 is cited when simply describing the observed results; it should be removed.

Response: Thanks. The Y axis showed the relative mRNA and protein expression of some barrier-related proteins, and therefore they bear no unit since it is a ratio. And we have removed reference 23 as suggested.

  1. The antibodies used should be better specified. Both JNK and ERK have doublets, are these antibodies specific for several isoforms? If so, this should be clearly indicated in the text and figures.

Response: Thanks. We have added the specificity information for JNK and ERK antibodies in the revised manuscript and also indicate the isoforms in the figures.

  1. The graphics are grainy, the quality of the images needs to be improved. Of all of them, for example, figures 7B and 7C are illegible.

Response: Thanks for pointing this out. We have improved the quality of the figures and enhanced the clarity of Figures 7B and C.

  1. In the results section, it would be interesting to include some p-values directly in the text to make the significance of the results more immediate.

Response: We have added p value in the results section as suggested.

  1. In the discussion, I would limit myself to discussing the results obtained and not to citing irrelevant studies as, for example, analyzing different compounds.

Response: We have deleted those irrelevant studies and replaced them with studies in which other benefical properties of gastrodin were examined.

  1. The discussion mentions how MPO promotes ROS production. Shouldn't their production be analyzed?

Response: Sorry we did not measure ROS in this study. We have modified this description and will take this into consideration in our future experimental design.

  1. The images provided for the blots are not exhaustive. it is necessary to provide the image of the entire gel without post production modifications in order to assess the antibody bands analyzed.

Response: Thank you for the nice comment, and we agree that generally the whole gel blot should be performed for the protein detection. In this case, most of the proteins we detected are well-characterzied proteins in different major signaling pathways, and these antibodies have been well characterized by many laboratories including us. Thus, for funding reason and to save the cost of antibodies, we cut the gel into sections based on the expected position of the target protein by referring to protein markers. Then we transferred each protein on each gel section to separate membranes and incubated the membranes with different antibodies, therefore we currently did not have a entire gel image with only one antibody.We will bear your suggestion in mind in our following studies to avoid this issue.

Reviewer 3 Report

Comments and Suggestions for Authors

Attached is the document with the revisions and suggestions.

Author Response

Comments:

The present article "Gastrodin alleviates DSS-induced colitis in mice through strengthening intestinal barrier and modulating gut microbiota" reported that gastrodin could alleviate DSS-induced colitis in a mouse model by enhancing gut barrier function, rectifying gut microbiota, and attenuating inflammatory responses. The findings suggest that gastrodin could effectively mitigate DSS-induced colitis, which could be achieved through enhancing the expression of mucins and tight junction proteins, alleviating oxidative stress, and suppressing the expression of pro-inflammatory cytokines through inflammation-related signaling pathways. Some adjustments and corrections are required as listed below:

  1. Introduction, line 54: Please include the molecular weight of gastrodin.

Response: We have added the molecular weight of 286 Da as suggested.

  1. Introduction, line 55: Cite the scientific name of the Gastrodia elata species here in the Introduction. Remember that when citing the scientific name of the species for the first time, indicate who described it, according to the rules of botanical nomenclature; remember that it should be in italics.

Response: Thanks for your suggestions. We have provided related information and iatlicized the botanical names.

  1. Materials and Methods, lines 74-75: Specify which protease inhibitors were used and their respective molecular weights. For example, was PMSF or EDTA used?

Response: Thanks for pointing this out. Acutally, we used a protease and phosphatase inhibitor cocktail from Roche, and we have added the product infomation in the revised method section.

  1. Materials and Methods, lines 74-75: Indicate the purity level of the analytically pure grade chemicals. Was it above 95%, above 99%? Include this information in the text.

Response: Thanks  we have added the information in the revised text.

  1. Materials and Methods, lines 123-127: Provide the percentage of powdered milk in the blocking solution and the dilutions of the primary and secondary antibodies.

Response: Thanks. We have provided the concentration of milk powder and dilutions of antibodies as suggested.

  1. Materials and Methods, line 123: Define TBST.

Response: We have expanded TBST as Tris buffered saline with Tween-20 when they first appear in the text.

  1. Materials and Methods, line 128: Provide the version and year of the ImageJ program used.

Response: We have provided the information as follows:

  1. Immunofluorescence assay, lines 120-134: Include the dilution of the antibodies.

Response: Thanks. The antibody dilution we used to perform immunofluorescence assay was 1:500, and we have added it in the manuscript.

  1. Statistical analysis: Specify which test was used to check the normality of the data. Remember that you can only use parametric tests if your data is normally distributed.

Response: Thanks for the reminder. We used Graphpad for statistical analysis and the normality of the data were checked by Shapiro-Wilk method before further analysis using parametric tests. We have added this information in the revised manuscript.

  1. Results, Figure 2: Improve Figure 2 by making the graphs larger. The current size makes it difficult for the reader to see the graphs clearly.

Response:Thanks. We have enlarged Figure 2 as suggested.

  1. Results, Figure 3: Improve the scale bar.

Response:Thanks. We have improved the scale bar as suggested.

  1. Results, Figure 4: Include the scale bar.

Response:Thanks. We have included a scale bar as suggested.

  1. Results, Figure 2E: The average colon length of a mouse of the strain used in the study is about 4 cm. How do you explain that the colon reached an average of 12 cm in the figure?

Response: Thanks Sorry for the confusion. Actually there are two types of units on the ruler shown in the Figure 2E. The unit on the right of the ruler is millimeter, which we used for measurement of the length of the colon. And 12 cm is among the normal range for the length of mouse colon.

  1. Results, Figure 7B: Enlarge the graph; the current size makes it very difficult for the reader to see clearly.

Response: Thanks. we have enlarged Figure 7B as suggested to make it more legible.

Round 2

Reviewer 2 Report

Comments and Suggestions for Authors

 I appreciate the changes made to the manuscript by the authors, however, I must insist regarding the blot images. Although the authors claim that most of the proteins detected are well-characterized proteins in several major signaling pathways and well-characterized by many laboratories, the presentation in this way of the membranes is not appreciable. it is important to point out that the membrane should be cut into several parts and not the gel as claimed by the authors. it is also important to note again that the molecular weights in this way are not visible so it is not possible to determine whether the antibodies are indeed those. I think the presence of the entire image of the membranes is very important.

Author Response

Comments: I appreciate the changes made to the manuscript by the authors, however, I must insist regarding the blot images. Although the authors claim that most of the proteins detected are well-characterized proteins in several major signaling pathways and well-characterized by many laboratories, the presentation in this way of the membranes is not appreciable. it is important to point out that the membrane should be cut into several parts and not the gel as claimed by the authors. it is also important to note again that the molecular weights in this way are not visible so it is not possible to determine whether the antibodies are indeed those. I think the presence of the entire image of the membranes is very important.

Responses: Thanks for your comments and suggestions. As suggested, we found some remaining samples and antibodies from freezers and ran Western blot for each protein on separate gels. And the images of the entire membrane (gel) are now provided to confirm the blot bands analyzed. We now assure that all the bands detected are at the expected sizes (molecular wights) along with the the molecular weight markers. We will also bear this in mind in our future work.
